# Yield, Characterization, and Possible Exploitation of *Cannabis Sativa* L. Roots Grown under Aeroponics Cultivation

**DOI:** 10.3390/molecules26164889

**Published:** 2021-08-12

**Authors:** Fabio Ferrini, Daniele Fraternale, Sabrina Donati Zeppa, Giancarlo Verardo, Andrea Gorassini, Vittoria Carrabs, Maria Cristina Albertini, Piero Sestili

**Affiliations:** 1Department of Biomolecular Sciences, University of Urbino Carlo Bo, Via Saffi 2, 61029 Urbino, Italy; f.ferrini2@campus.uniurb.it (F.F.); daniele.fraternale@uniurb.it (D.F.); v.carrabs@campus.uniurb.it (V.C.); maria.albertini@uniurb.it (M.C.A.); piero.sestili@uniurb.it (P.S.); 2Department of Agricultural, Food, Environmental and Animal Sciences, University of Udine, 33100 Udine, Italy; giancarlo.verardo@uniud.it; 3Department of Humanities and Cultural Heritage, University of Udine, 33100 Udine, Italy; andrea.gorassini@uniud.it

**Keywords:** *Cannabis sativa* L., aeroponics, roots, campesterol, stigmasterol, β-sitosterol, epi-friedelanol, friedelin

## Abstract

*Cannabis sativa* L. has been used for a long time to obtain food, fiber, and as a medicinal and psychoactive plant. Today, the nutraceutical potential of *C.*
*sativa* is being increasingly reappraised; however, *C. sativa* roots remain poorly studied, despite citations in the scientific literature. In this direction, we identified and quantified the presence of valuable bioactives (namely, β-sitosterol, stigmasterol, campesterol, friedelin, and epi-friedelanol) in the root extracts of *C. sativa,* a finding which might pave the way to the exploitation of the therapeutic potential of all parts of the *C. sativa* plant. To facilitate root harvesting and processing, aeroponic (AP) and aeroponic-elicited cultures (AEP) were established and compared to soil-cultivated plants (SP). Interestingly, considerably increased plant growth—particularly of the roots—and a significant increase (up to 20-fold in the case of β-sitosterol) in the total content of the aforementioned roots’ bioactive molecules were observed in AP and AEP. In conclusion, aeroponics, an easy, standardized, contaminant-free cultivation technique, facilitates the harvesting/processing of roots along with a greater production of their secondary bioactive metabolites, which could be utilized in the formulation of health-promoting and health-care products.

## 1. Introduction

Modern nutrition increasingly involves the consumption of functional foods and nutraceuticals that are useful for health maintenance, and for the prevention and treatment of some diseases [1].

Functional foods are solid and/or liquid foods, processed or not, which in addition to nourishing also contain biologically active compounds associated with health benefits. Nutraceuticals—the technical evolution of functional foods—are preparations such as tablets, syrups, powder, etc. that contain active ingredients mostly extracted from plant foods (botanicals); nutraceuticals must provide clinically proven health benefits and can be used to prevent, manage, and treat certain chronic diseases [2].

In parallel, the use of health-promoting products containing the above bioactives for non-nutritional purposes (skin care, inhalers for the respiratory tract, rhinological and otological formulations, eyedrops) is on the rise [3].

*Cannabis sativa* L. is a dioecious annual herbaceous plant, also defined as a big grass, native to Central Asia and belonging to the botanical family Cannabaceae. For centuries, it has been essential for humans for food, fiber, and as a medicinal and psychoactive plant. Today, the nutraceutical potential of *C. sativa* is being increasingly reappraised [4].

Two subspecies can be identified within *C. sativa*: *C. sativa* subsp. *indica*, which contains more than 20% of the psychoactive compound D9-tetrahydrocannabinol (THC) in the resin produced by the female buds; and *C. sativa* subsp. *sativa,* which contains numerous bioactive molecules in all parts of the plant, but shows a much lower content of psychoactive molecules, particularly THC, which must not exceed 0.2 % [5].

The subspecies *sativa* is now commonly and legally cultivated in many European countries for the production of cannabidiol/cannabidiolic acid (CBDs)—the most abundant non-psychoactive cannabinoid—and seeds, which represent an excellent source of nutrients, such as lipids, proteins, carbohydrates, minerals, vitamins, amino acids, essential fatty acids, and insoluble fiber. Edible oil and flour for human use can be obtained from *C. sativa* seeds [6]. It has been shown that even the sprouts obtained from the germination of *C. sativa* seeds represent an interesting example of a functional food, because sprouts after three/five days are rich in compounds with anti-inflammatory activity, such as the prenylflavonoids cannaflavins A and B [5].

Interestingly, in the 1970s, Slatkin et al. (1971) [7] and Sethi et al. (1977) [8] demonstrated the presence of triterpenoids and sterols in the root extracts of mature *C. sativa* plants. More recently, Jin Dan et al. (2020) [9] profiled the groups of secondary metabolites in the individual parts of the plant: the authors highlighted the presence of nutraceutically relevant constituents, namely, sterols (β-sitosterol, stigmasterol, campesterol) and triterpenoids (friedelin and epi-friedelanol) in mature *C. sativa* roots, paving the way to the reappraisal and implementation of the therapeutic potential of all parts of the *C. sativa* plant.

Indeed, although historically *C. sativa* roots were widely used as medicines to treat inflammatory conditions, joint pain, gout, and more, their therapeutic potential has been largely ignored in modern times. To our best knowledge, few studies have examined the composition of *C. sativa* roots and their medical potential [10], and even less have explored alternative methods for the production of *C. sativa* roots or to increase their content of biologically active molecules.

The relative scarcity of similar studies likely depends on the fact that the research on *C. sativa* is mainly focused on the most widespread cannabinoids, which are not significantly present in the roots [10].

With the aim of filling this gap, in the present work, *C. sativa* var. *Kompolti*—a legal variety routinely used for food production purposes—was cultivated through aeroponics [11], a method that allows the plants to grow in a highly controlled manner, free of contaminants and suspended in an environment devoid of soil or other means of support. This system was selected among others as it theoretically allows a greater production of roots as compared to traditional cultivation in soil, with the additional advantage that there is no need for time-consuming and expensive rinsing procedures to isolate and process the roots.

As a consequence, the aeroponics culture should provide a greater production of *C. sativa* roots and a higher yield of secondary bioactive metabolites.

To test this hypothesis, here, we characterized and compared the morphological features of plants grown in aeroponics (AP) or in soil (SP), and the secondary metabolites contained in their roots, with the aim of determining the yield of nutraceutically relevant compounds, which could be included as active ingredients in the formulation of health-promoting and health-care products.

## 2. Results

### 2.1. Biomass Production

The growth of *C. sativa* var. Kompolti plants used in this study was significantly influenced by the two systems tested, i.e., conventional vs. aeroponic, with aeroponics promoting a significantly more rapid and intense growth of both the aerial parts and root systems (Figure 1; Table 1). On average, after 8 weeks of parallel cultivation, the roots of APs showed a 64-fold and 13-fold higher fresh (FW) and dry weight (DW) as compared to SP, respectively; the aerial parts showed a 39-fold and 44-fold higher FW and DW; the stems’ average diameter and the mean leaves area increased by 3.89-fold and 8.9-fold, respectively. AP and AEP reached almost double the height (ca. 70 cm) as compared to SP (ca. 30 cm). Finally, the addition of the elicitor salicylic acid to the nutrient spray did not result in any significant variation in these parameters in AEP as compared to AP (Table 1, Figure 1).

### 2.2. Extract Characterization

The content of the main roots’ bioactive constituents was comparatively determined in SP, AP, and AEP plants by GC-MS. The main compounds identified were the phytosterols campesterol, stigmasterol, and β-sitosterol, and the triterpenes epi-friedelanol and friedelin (Figure 2). Figure 3 shows a typical gas chromatogram of the ethyl acetate extract of powdered *C. Sativa* roots. On a per DW basis (Table 1; Figure 4), the amount of bioactives was higher in SP as compared to both AP and AEP. Regarding the single constituents, the amount of epi-friedelanol and friedelin was far higher in SP, that of campesterol and stigmasterol was similar in the three types of cultures, while β-sitosterol was higher in AP and AEP. On a percent basis (Table 2 and Figure 5), friedelin and epi-friedelanol were the most expressed compounds in SP, while β-sitosterol was the most expressed in AP and AEP; finally, the amount of β-sitosterol decreased and the amount of epi-friedelanol increased in AEP as compared to AP. No other bioactive metabolite (i.e., monoterpenes, sesquiterpenes, or cannabinoids) was detected.

Notably, when analyzed on a “per plant” basis the results were markedly different (Figure 6). Indeed, since the biomasses of the plants grown in aeroponics were heavier (13-fold to 64-fold for DW and FW roots, respectively), both AP and AEP contained significantly higher amounts of root bioactives (Table 3; Figure 6). The amounts of β-sitosterol from AP (10.86 ± 0.72 mg) and AEP (9.89 ± 2.17 mg) roots were 23 and 20 times higher than in SP (0.49 ± 0.05mg), respectively; friedelin, whose concentration on a per weight basis was significantly higher in SP, exhibited higher per plant values in AEP and AP (4.55 ± 0.47 mg; 5.67 ± 0.4 mg, respectively) than SP (2.37 ± 0.3 mg) (Table 3; Figure 6). Similar proportions were observed for campesterol, stigmasterol, and epi-friedelanol (Table 3).

## 3. Materials and Methods

### 3.1. Chemicals and Reagents

Extraction solvents (analytical grade), cholesterol, β-sitosterol, stigmasterol, campesterol, friedelin, and saturated *n*-alkanes standard (C7–C40) were obtained from Sigma-Aldrich (St. Louis, MO, USA).

### 3.2. Plant Material and Cultures

*C. sativa* Kompolti seeds were supplied by Appennino Farm, Gaggio Montano, Bologna (Italy), lot B30756201900001. The seeds were germinated in filter paper wetted with distilled water—10 seeds in a 14 cm diameter glass Petri dish—in the dark, at a constant temperature of 25 °C. After 4 days, the rooted seeds were transferred into plastic pots, with a diameter of 5.5 cm and a height of 6.0 cm, containing a mixture of 50% peat and 50% vermiculite wetted with Hoagland’s half-strength nutrient solution up to when the seedlings were ready for transplanting. The pots (one rooted seed per pot) were placed in a climatic cell with a photoperiod of 18 h (lamp and conditions as below) until the first two true leaves were fully developed. At this point, five *C. sativa* seedlings of uniform size, the first two true leaves being equal to about 3.0 cm, were selected each time. The base of the stem of each seedling was fixed with a sponge, placed in a reticulated pot for aeroponics (with a diameter of 6.0 cm and a height of 6.0 cm), and this was placed on the hole of the lid of a tub for aeroponic cultivation in black PVC (50.0 cm × 50.0 cm × 34.0 cm in height), capable of hosting five plants each time, at a distance of 15 cm from each other. Complete Hoagland’s nutrient solution was sprayed at the roots of the plants for a duration of 15 min every hour, and recovered via a closed circuit. The electrical conductivity (EC) and pH of the nutrient solution were controlled at 0.6–0.7 ms/cm and 6.0, respectively. The aeroponics culture system was maintained in a climatic cell with a photoperiod as stated above with high pressure sodium lamps (Sonlight AGRO 250W grown + bloom). The photosynthetic photon flux density (PPFD) at the plant canopy was about 150 µmol/m^2^/s. Temperatures during the periods of light and dark were maintained at 27 ± 1 °C and 22 ± 1 °C, respectively. The relative humidity was 65 ± 5% and the mean CO_2_ concentration was 670 ± 30 µmol/mol.

For the traditional cultivation in pots, the seedlings with two real leaves (about 3 cm long) were placed in a plastic pot (with a diameter of 30.0 cm and a height of 30.0 cm), containing a mixture as stated above (50% vermiculite and 50% peat), three plants per pot, spaced 15 cm from each other, and watered periodically with whole Hoagland’s nutrient solution. The pots were placed in a climatic cell with the conditions described for aeroponics and at the same time as this.

Three culture systems of *C. sativa* Kompolti were set up: SP (soil plant), carried out only for the vegetative phase; AP (aeroponic plant), carried out only for the vegetative phase; AEP (aeroponic-elicited plant), carried out only for the vegetative phase supplemented with the elicitor technique with salicylic acid.

AEPs were obtained by adding salicylic (25 µM final concentration) acid to the nutrient solution of one-week old plants, according to Ze-Bo Liu et al. [12].

### 3.3. Biomass Production

Five plants from each culture system were harvested three times after 8 weeks of culture. It should be noted that the plants were kept in the vegetative phase by maintaining the photoperiod constant (18 h). Under these conditions, plants could not flower because flowering requires a gradual reduction of the photoperiod (from 18 to 12 h). The root systems were carefully washed with tap water, dried with absorbent paper to remove excess water, and their fresh weights were immediately recorded. The other parameters taken into consideration were as follows: dry weight of the roots (g); fresh and dry weight of the aerial parts (g); height of the aerial parts (cm); diameter of the stems (mm); and surface of the collected leaves at half height (cm).

### 3.4. Extraction of Dry C. Sativa Roots

Powdered *C. sativa* roots (300.0 mg) and 100 µL of the IS solution (cholesterol, 1.128 mg/mL in ethyl acetate) was extracted in ethyl acetate (40 mL) under magnetic stirring for 1.5 h at room temperature, followed by centrifugation at 5000 rpm for 8 min. The supernatant was collected in a flask and the residue was extracted once again in the same manner. The collected organic phases were washed with water (2 × 10 mL) and brine (10 mL) then dried (Na_2_SO_4_ anhydrous), filtered, and evaporated to dryness in vacuo at 30 °C. The residue was dissolved in 10 mL of ethyl acetate and kept at 4 °C until GC-MS and GC-FID analyses.

### 3.5. Gas Chromatography (GC-MS, GC-FID)

GC-MS analyses were carried out using a Trace GC Ultra gas chromatograph coupled to an ion-trap mass spectrometer (ITMS) detector Polaris Q (Thermo Fisher Scientific, Italy) and equipped with a split–splitless injector. The column was a 30 m × 0.25 mm i.d., with a 0.1 µm film thickness, and a fused silica SLB-5ms (Supelco, Sigma-Aldich, Italy). The initial oven temperature was 240 °C programmed to 280 °C at 2 °C/min and kept at 280 °C for 5 min; the temperature was then raised to 310 °C at a rate of 10 °C/min and maintained at this temperature for 7 min. Samples (1 µL) were injected in the split (1:10) mode. The injector, transfer line, and ion source were set at 280, 280, and 200 °C, respectively. Helium was used as carrier gas at a flow of 1 mL min^−1^. The mass spectra were recorded in electron ionization (EI) mode at 70 eV electron energy with a mass range from *m/z* 50 to 650 and a scan rate of 0.8 scan/sec. Identification of metabolites was carried out by comparing the spectral data and retention times with the standards and spectra from the NIST02 spectral library.

A Fisons GC 8000 series gas chromatograph, equipped with a flame ionization detector and a split–splitless injector (Fisons Instruments, Milan, Italy), was used for the quantitation of secondary metabolites. The separation was carried out with a fused silica capillary column DB-5MS UI of 30 m × 0.250 mm × 0.25 µm film thickness (Agilent, J&W, Italy). The initial oven temperature was 240 °C programmed to 280 °C at 2 °C/min and kept at 280 °C for 10 min; the temperature was then raised to 310 °C at a rate of 10 °C/min and maintained at this temperature for 15 min. Samples (1 µL) were injected in the split (1:10) mode. The injector and detector were set at 280 °C. Hydrogen was used as carrier gas at a flow of 1.8 mL/min. Peak areas were integrated using a Varian Galaxie Workstation (Agilent Technologies, Cernusco sul Naviglio, Italy).

Quantification of the analytes in the dry *C. sativa* roots was performed using the internal standard method based on the relative peak area of analyte to IS (cholesterol) from the average of three replicate measurements. When standards were unavailable, the quantification of the target analyte was carried out using the relative response factor of the available standards of similar chemical structure.

The retention indices (RIs) were calculated by comparing the retention time of each compound with those of a homologous series of *n*–alkanes standard (C7–C40) under the same chromatographic conditions.

### 3.6. Statistics

Statistical analyses were carried out to compare each root metabolite quantified in AP AEP and SP using Tukey’s multiple comparison tests.

## 4. Discussion

This study was undertaken because *C. sativa* roots—although potentially valuable—are a neglected source of bioactive compounds. The fact that the roots lack the most pharmacologically relevant compounds, namely, THC and CBDs, and the relative complexity of root processing, probably accounts for the lack of interest in *C. sativa* roots. However, unlike SP, root harvesting/processing from AP is far easier, cheaper, and more flexible. Indeed, AP roots are clean and free from the parasites and contaminants normally present in the soil. Moreover, interestingly, as shown in the present study and elsewhere [13,14], the absolute and relative yield of bioactives can be modulated by simply varying the composition of the nutrient sprayed onto the roots by adding specific elicitors. Furthermore, AP roots may meet organic cultivation standards, which today represent a desired and high-quality reference point.

The chemical characterization of SP, AP, and AEP roots indicates that: (1) the phytosterols β-sitosterol, campesterol, and stigmasterol, and the triterpenoids friedelin and epi-friedelanol represent the major components of interest; (2) the relative proportion of these constituents was significantly affected by the cultivation system.

Here, we demonstrated that aeroponics applied to *C. sativa* var. Kompolti resulted in a significant modification in the yield of plant biomasses and in the net and relative abundance of root bioactive compounds, as compared to conventional soil cultivation. Indeed, the biomass of AP (both the aerial parts and roots) after 8 weeks of cultivation strikingly outpaced that of SP. In particular, root growth was impressive with a 64-fold and 13-fold increase in FW and DW, respectively, as compared to the roots from SP. The greater increase in FW was likely due to the very high hydration rate attainable under aeroponics.

This growth was also found to occur in other plants: Li et al. [15] showed that the root DW of two varieties of lettuce grown in aeroponics was significantly higher than that obtained both through cultivating plants in soil and with the hydroponic technique.

Other authors [16] showed that aeroponics used to produce *Crocus sativus* (saffron) promoted more robust growth of the root system, which was not paralleled by a proportional growth of the aerial parts. This observation suggests that, under aeroponics, a larger root system does not necessarily result in a correspondingly greater biomass of the aerial parts. We also found that, although both the roots and aerial parts of AP were invariably greater and heavier than those of SP, the aerial parts and roots grew to different extents. The differential growth observed here and elsewhere may be the expression of a plant type-dependent effect of aeroponic cultivation. In fact, both the traditional cultivation method in substrate and the hydroponic method involve the continuous immersion of the roots in nutrients and water. In some cases, this can more efficiently stimulate the growth of the aerial part of the plant as compared to the aeroponics method, in which the roots are suspended in a chamber, wetted at regular intervals with the nutrient solution, and have virtually unlimited access to oxygen. The extensive oxygen availability likely represents the most important advantage of the aeroponic culture method over conventional and hydroponic ones [17]. 

From a pharmaceutical perspective, Hayden et al. [18,19] demonstrated that aeroponics represent an excellent system for the production of roots from medicinal plants. These organs can then be used for the extraction of active molecules, such as, for example, from burdock (*Arctium lappa*) and ginger (*Zingiber officinale*). Importantly, as compared to other techniques, aeroponics allows perfectly clean root apparatuses to be obtained, which can be immediately harvested, extracted, lyophilized, micronized, or subjected to other processing methods. A further advantage of aeroponics is that the culture medium can be easily and precisely enriched with specific elicitors, which can further enhance the yield of bioactives.

Although more appreciated for its CBD content from the plant’s inflorescences and for the wide utilization of the aerial parts, here, we applied and tested aeroponics to produce *C. sativa,* the roots of which have a fairly high content of potentially valuable constituents characterized by attractive pharmacological, nutraceutical, and cosmeceutical activities. Notably, we show that the yield of these components on a per plant basis was invariably and significantly higher in AP and AEP as compared to SP, and, by using salicylate as elicitor [20,21], the accumulation of all bioactives could be achieved (with the exception of β-sitosterol, which only slightly decreased as compared to AP).

Hence, these findings could pave the way for the rational implementation of *C. sativa* aeroponic cultivation based on the development of mixtures of specific elicitors to increase and optimize the yield of root bioactive constituents. In light of this, elicitation may allow different *C. sativa* roots’ phytocomplexes to be obtained with specific attitudes resulting from the relative proportion and peculiar biological features of each component.

As briefly anticipated above, the properties of the bioactives identified in *C. sativa* roots largely justify their use—either singularly or as phytocomplex—for the preparation of health-promoting products. The most abundant components are represented by phytosterols, particularly β-sitosterol. β-Sitosterol is a sterol found in almost all plants. It is one of the main subcomponents of a group of plant sterols known as phytosterols, which are very similar in composition to cholesterol. High levels are found in rice bran, wheat germ, corn oil, and soybeans; peanuts and their products such as peanut oil, peanut butter, and peanut flour; *Serenoa repens,* avocados, pumpkin seed, *Pygeum africanum,* and cashew fruit.

The antihypercholesterolemic effect of β-sitosterol was reported by Cicero et al. [22]; thirty-six human volunteers took 2g/day of β-sitosterol and 8g/day of soy protein for 40 days and after this period they showed a significant decrease in LDL (low-density lipoprotein), VLDL (very low density lipoprotein), and TG (triglycerides) levels, and a significant increase in HDL (high-density lipoprotein) [22].

Studies suggest that β-sitosterol inhibits the proliferation of human prostate cancer cells [23] and the growth of tumors derived from PC-3 human prostate cancer cells [24,25].

The ability of β-sitosterol and other phytosterols to inhibit aromatase and 5-alpha-reductase has been well documented, and this inhibitory capacity has been exploited to treat pathologies such as benign prostatic hyperplasia and androgenetic alopecia [26,27,28]. One of the richest sources of phytosterols, particularly β-sitosterol, is *Serenoa repens,* whose extracts have been largely studied and used in nutraceutical formulations (Permixon, Calprost, Difaprost) proposed for the adjunct therapy of benign prostatic hyperplasia [25,29,30].

The concentration of β-sitosterol in *S. repens* extract, 0.454 ± 0.018mg/g, dry mass [31], is a hundred times lower than that found in *C. sativa* extracts. Such a finding would make the use of *C. sativa* root extracts for nutraceutical applications as plausible as for *S. Repens*. As compared to other sources, *C. sativa* roots have a very low lipid, proteins, and carbohydrates contents; hence, they could be taken by overweight/obese, diabetic, or hypercolestolemic patients without increasing their caloric intake.

Friedelin, a pentacyclic triterpenoid which can be found in many plants, displays a wide spectrum of anti-inflammatory, antipyretic, anticarcinogenic, and antitumor effects, with a low toxicity [32].

Friedelin was reported to promote apoptosis and inhibit the growth of various tumor cell lines including MCF-7 human breast cancer and AML-196 human leukemia cells [33,34,35,36].

Friedelin also possesses other remarkable properties, such as in vivo anti-inflammatory, analgesic, and antipyretic effects in adult Wistar rats [37]; mast cell membrane stabilization [38]; a hypoglycemic effect in diabetic rats [39]; gastroprotective and antiulcerogenic activity [40]; estrogenic activity [41]; and antihyperlipidemic and antihypertensive effects [42].

Friedelin exhibits a remarkable antioxidant capacity, comparable to that of BHT or ascorbate [43].

Finally, friedelin, thanks to its antimycobacterial activity, has been proposed as a natural antituberculosis agent [44]. Interestingly, this usage parallels that of *C. sativa* leaf, macerated in warm water and taken as a treatment for tuberculosis by the Bapedi healers of Limpopo Province, South Africa [45]. 

Another molecule present in significant percentages is epi-friedelanol (Figure 3). This compound is another pentacyclic triterpenoid that shares a large part of its molecular structure with friedelin, with the difference that the latter has a cyclohexanone, while epi-friedelanol has a cyclohexanol.

Epi-friedelanol is present in several plants, such as in the root barks of *Ulmus Davidiana* [46], *Cayratia trifolia* [47,48,49], *Vitis trifolia* [50], *Celtis sinensis* [51], *Mallotus apelta* [52], and *Ulmus pumila* [53]. This triterpenoid has been reported to have anticancer [50,54], anti-inflammatory [51], and anti-senescence activity [46]. In particular, Yang et al. [46], who found that epi-friedelanol suppresses cellular and replicative senescence in human fibroblasts and human umbilical vein endothelial cells, proposed its use in the formulation of nutraceuticals or cosmeticeutics aimed at modulating tissue aging or aging-associated diseases.

## 5. Conclusions

In summary, the main constituents of *C. sativa* roots, namely, β-sitosterol, friedelin, and epi-friedelanol, possess converging or complementary biological activities in such a way that their co-presence in *C. sativa* root extracts may result in additive or even synergistic effects, which could be used as adjunctive treatment in several pathological and physiopathological conditions, such as inflammatory states, dyslipidemias, hyperglycaemia, menopause, and skin aging.

Hence, the aforementioned considerations make *C. sativa* roots obtained through aeroponic cultivation a valuable material. This is further emphasized by the striking biomass yield, the high bioactive content, and the ease of root harvesting/processing associated with this technique.

## Figures and Tables

**Figure 1 molecules-26-04889-f001:**
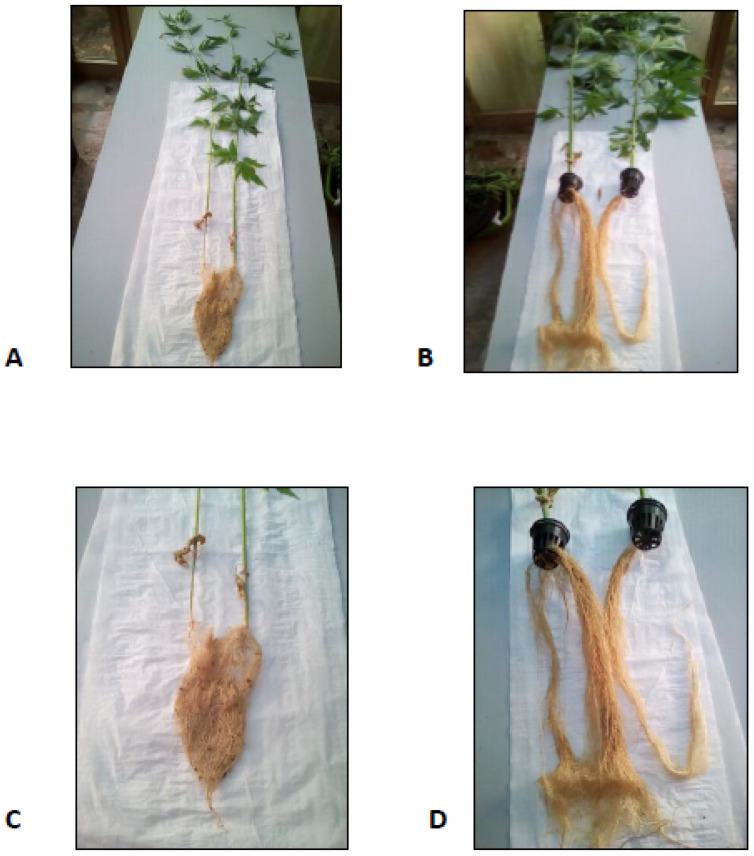
*C. sativa* L. plants and roots cultured either under soil (**A**,**C**) or aeroponic conditions (**B**,**D**).

**Figure 2 molecules-26-04889-f002:**
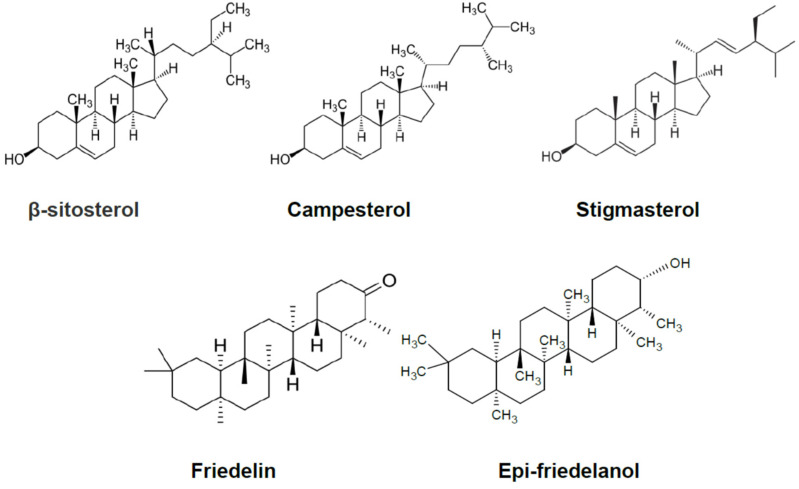
Chemical structure of the main chemical compounds identified by GC-MS in aeroponic plant (AP), aeroponic-elicited plant (AEP), and soil-cultivated plant (SP) root extracts.

**Figure 3 molecules-26-04889-f003:**
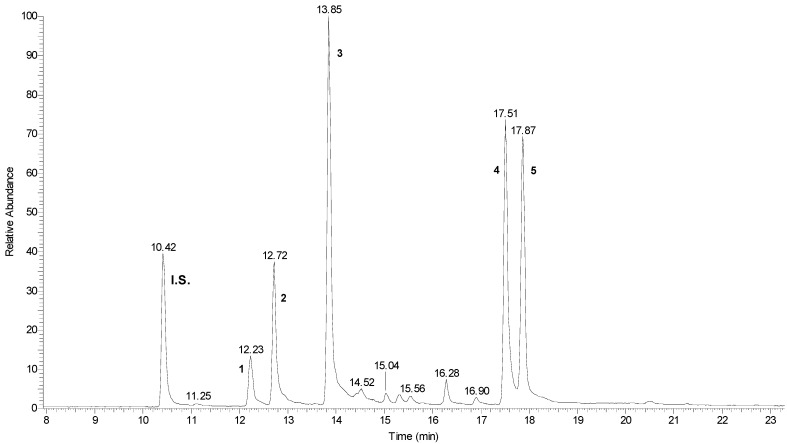
Typical total ion current (TIC) chromatogram obtained from the ethyl acetate extract of powdered *C. Sativa* roots. Peaks: **I.S.** (internal standard, cholesterol); **1** (campesterol); **2** (stigmasterol); **3** (β-sitosterol); **4** (epi-friedelanol); **5** (friedelin).

**Figure 4 molecules-26-04889-f004:**
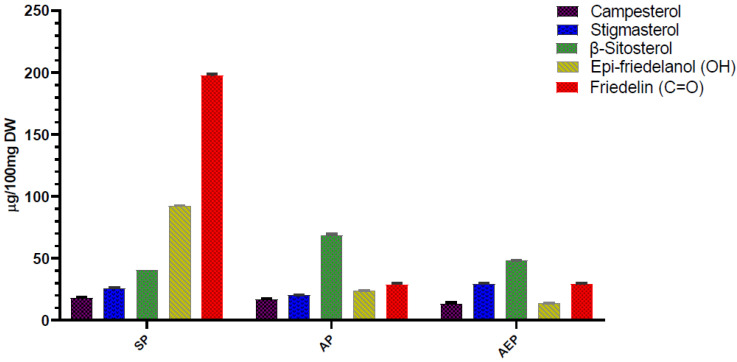
Concentration of the main bioactive compounds of aeroponic plant (AP), aeroponic-elicited plant (AEP), and soil-cultivated plant (SP) root extracts of *C. sativa* L. Data are expressed as µg/100mg DW ± SEM.

**Figure 5 molecules-26-04889-f005:**
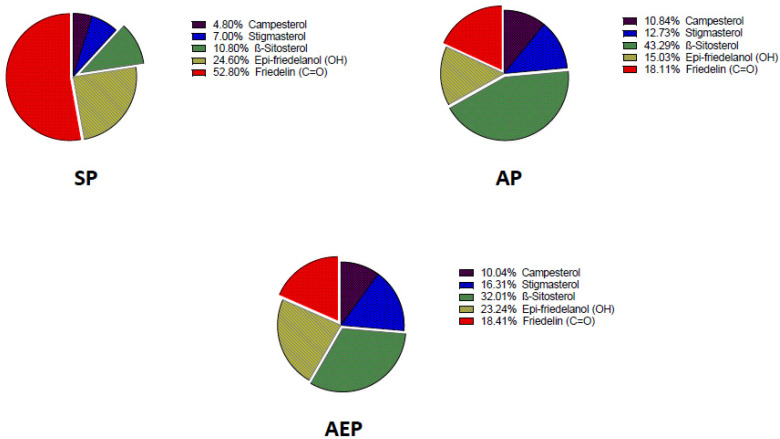
Percentage of the main bioactive compounds of the *C. sativa* L. root extracts of aeroponic plant (AP), aeroponic-elicited plant (AEP), and soil-cultivated plant (SP).

**Figure 6 molecules-26-04889-f006:**
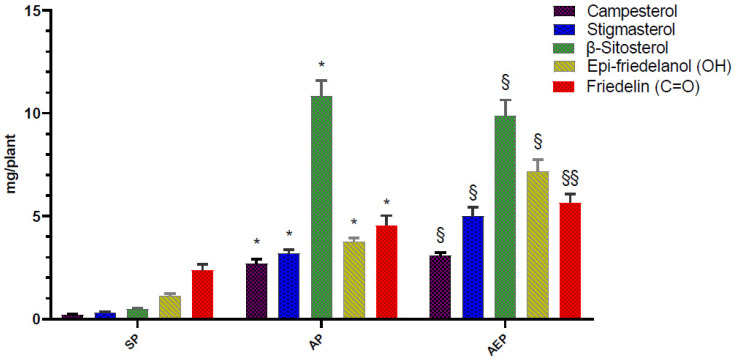
Total amount of the main bioactive compounds of aeroponic plant (AP), aeroponic-elicited plant (AEP), and soil-cultivated Plant (SP) *C. sativa* roots. Data are expressed as mg per plant. * *p* <.0001 AP vs. SP; §§ *p* < 0001 and § *p* < 05 AEP vs. AP (Tukey’s multiple comparisons test).

**Table 1 molecules-26-04889-t001:** Morphological Features of different plants.

*Culture System*	Height (cm)	Roots Weight FW/DW (g)	Aerial Parts Weight FW/DW (g)	Stem Average Diameter (mm)	Leaves Average Area (cm^2^)
**SP**	32.6 ± 0.7	3.7 ± 0.4/1.2 ± 0.1	15.8 ± 0.7/2.9 ± 0.4	2.13 ± 0.1	5.2 ± 0.4
**AP**	70.5 ± 1.8	238.7 ± 4.1/15.8 ± 0.5	616.9 ± 3.0/129.3 ± 0.7	8.30 ± 0.5	48.3 ± 3.7
**AEP**	72.5 ± 1.3	246.1 ± 4.3/16.7 ± 0.4	656.3 ± 3.1/137.1 ± 1.6	8.21 ± 0.4	49.4 ± 3.4

**Table 2 molecules-26-04889-t002:** Content of the main bioactive compounds present in 100.0 mg of dry SP, AP, and AEP roots.

		Culture System					
	RI	SP		AP		AEP	
Compound		**(µg) ^a^**	**%**	**(µg) ^a^**	**%**	**(µg) ^a^**	**%**
Campesterol	3179	17.8 ± 0.6	4.8	17.2 ± 0.4	10.9	18.5 ± 0.2	10
Stigmasterol	3205	26.1 ± 0.2	7	20.2 ± 0.2	12.7	30.1 ± 1.0	16.3
β-sitosterol	3263	40.4 ± 0.0	10.8	68.7 ± 1.1	43.3	59.0 ± 7.8	32
Epi-friedelanol	3433	92.2 ± 0.5	24.6	23.9 ± 0.1	15	42.8 ± 1.5	23.2
Friedelin	3448	197.5 ± 1.3	52.8	28.8 ± 1.2	18.1	33.9 ± 1.0	18.5
Total		374.0 ± 2.6		158.8 ± 1.9		184.3 ± 8.5	

^a^ Data are expressed as the mean value ± standard deviation; *n* = 3 repetitions.

**Table 3 molecules-26-04889-t003:** Total amount of chemical compounds in SP, AP and AEP cultivation.

	Total mg/Plant	Total mg/Plant	Total mg/Plant
**Campeterol**	0.21 ± 0.03 mg	2.72 ± 0.2mg	3.09 ± 0.14 mg
**Stigmasterol**	0.31 ± 0.04 mg	3.19 ± 0.17 mg	5.03 ± 0.4 mg
**β-Sitosterol**	0.49 ± 0.05mg	10.86 ± 0.72 mg	9.9 ± 2.17 mg
**Epi-friedelanol (OH)**	1.11 ± 0.14mg	3.77 ± 0.17 mg	7.16 ± 0.6 mg
**Friedelin (C=O)**	2.37 ± 0.3mg	4.55 ± 0.47 mg	5.67 ± 0.4mg

## Data Availability

Raw data supporting reported results can be requested to the corresponding author.

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
