# Peer review of "Yield, Characterization, and Possible Exploitation of Cannabis Sativa L. Roots Grown under Aeroponics Cultivation"

_molecules, 2021, doi:10.3390/molecules26164889_

Round 1
Reviewer 1 Report
Manuscript molecules-1296110 presents the sterol/triterpenoid profiles of Cannabis sativa roots obtained using different culture methods. The authors present evidence that aeroponic cultivation of C. sativa yields roots that may be of nutraceutical importance.
There are some suggestions and corrections for the authors to address:
Abstract, line 1 and throughout the manuscript: “Sativa” should not be capitalized.
Abstract, line 11: free-of-contaminant [add hyphens].
Table 2: Please determine and include retention indices (RI) in the table. This will be useful to future investigators. Were any monoterpenes, sesquiterpenes, or cannabinoids detected? The total % from Table 2 = 100% for each culture system. Does this imply that no other metabolites were detected?
Please include a representative gas chromatogram as a figure.
Figure 2: It would be useful to place the structure of friedelin in the same orientation as that for epi-friedelinol.
Page 10, paragraph 5: Other authors… [“authors” not capitalized].
Page 11, paragraph 3: thirty-six human volunteers… [insert hyphen].
Author Response
All the points raised by Referee #1 have been addressed.
In particular:
a representative gas chromatogram has been included as a figure (Fig. 3);
Fig. 2 has been modified to present the structure of friedelin in the same orientation as that for epi-friedelinol.
The retention indices (RI) have been included in table 2.
The absence of other relevant metabolites is now specified in the text (paragraph 3.2).

Reviewer 2 Report
The manuscript of molecules-1296110, Yield, characterization and possible exploitation of Cannabis Sativa L. roots grown under aeroponics cultivation, presents a work on the content of bioactive substances (namely β-sitosterol, stigmasterol, campesterol, friedelin and epi-friedelanol) in Cannabis Sativa L. root extracts. The work contains all the elements of the research work, but in my opinion is is lacking some elements.
At First, must be som information on why these ingredients were studied? What is the purpose of this? Because at first glance, the breeding technique is more exposed than the fact why it was done.
Secondly, how did the content of the tested ingredients change over time and what would the positive / negative impact of this plant have on the values ​​of this plant?
Author Response
At First, must be som information on why these ingredients were studied? What is the purpose of this? Because at first glance, the breeding technique is more exposed than the fact why it was done.
Agreed. To better clarify the purpose of the study the Introduction has been changed as follows.
"Jin Dan et al., (2020) [9] profiled the groups of secondary metabolites in the individual parts of the plant: the authors highlighted the presence of nutraceutically relevant constituents, namely sterols (β-sitosterol, stigmasterol, campesterol) and triterpenoids (Friedelin and Epi-friedelanol) in mature C. Sativa roots, paving the way to the reappraisal and implementation of the therapeutic potential of C. Sativa in all its parts.
Indeed, although historically C. Sativa roots were widely used as medicines to treat inflammation, joint pain, gout and more, the therapeutic potential of C. Sativa roots has been largely ignored in modern times. To our best knowledge few studies have examined the composition of C. Sativa roots and their medical potential [10], and even less are those in which alternative methods are considered for increasing the production of the roots and their content of biological active molecules.
The relative scarcity of such studies likely depends on the fact that the research on C. Sativa is mainly focused on the most widespread cannabinoids that are not significantly present in the roots [10].
With the aim of filling this gap, in the present work C. sativa subsp. sativa var. Kompolti - a legal variety routinely used also for food production purposes - has been cultivated through “aeroponic culture” (AP) [11] , in which the plants grow in a highly controlled manner, free of contaminants and suspended in an environment devoid of soil or other support means. This system has been selected among others as it theoretically allows a greater production of roots as compared to traditional cultivation in soil (SP), with the additional advantage that there is no need for time consuming and expensive rinsing procedures to isolate and process the roots.
As a consequence the aeroponics culture should provide a greater production of C. Sativa roots and a higher yield of secondary bioactive metabolites.
To test this hypothesis, here we have characterized and compared AP and SP morphological features as well as the secondary metabolites contained in their roots with the aim of determining the yield of nutraceutically relevant compounds which could be included as active ingredients in the formulation of health promoting and health care products."
Secondly, how did the content of the tested ingredients change over time and what would the positive / negative impact of this plant have on the values ​​of this plant?
Good point. The main goal of this paper is to evaluate the effect of aeroponic culture over the traditional culture in Canapa sativa and characterize the amount of each relevant compound in roots before inflorescence. Ongoing studies by our group are being focused on the bioactive constituents profile (in either roots and aerial parts) in different phases of growth of aeroponic vs soil cultivated Canapa plants.
Round 2
Reviewer 2 Report
Now it's Ok.
This manuscript is a resubmission of an earlier submission. The following is a list of the peer review reports and author responses from that submission.